# Identification of personal factors that influence engagement in cardiac rehabilitation and interventions targeting personal factors: A scoping review protocol

**Hongyu Zhang**◯*, **Duygu Sezgin**◯

School of Nursing and Midwifery, University of Galway, Galway, Ireland

* H.Zhang15@universityofgalway.ie

## Abstract

Cardiac rehabilitation effectively reduces mortality and enhances the quality of life for individuals with cardiovascular disease. Despite that, individuals' engagement in cardiac rehabilitation remains low. Considering the significant contributions of individuals' self-management of cardiovascular disease to their progress, it is essential to understand the personal factors that influence engagement in cardiac rehabilitation. This scoping review aims to identify and map personal factors that influence cardiac rehabilitation engagement with a specific focus on the subjective experiential dimensions of personal factors (cognitive, emotional, and behavioural). It also aims to explore interventions targeting personal factors to increase cardiac rehabilitation engagement. This review will be reported using the PRISMA-ScR checklist following the Joanna Briggs Institute (JBI) methodology. It will include peer-reviewed articles published in English from January 2004, excluding grey literature. Studies reporting adult populations aged 18 and over with cardiovascular disease and addressing personal factors or interventions to increase cardiac rehabilitation engagement, will be included. Databases for the searches will include PubMed, Embase, Cochrane Library, Cumulative Index to Nursing and Allied Health Literature, PsycINFO, Scopus, and Web of Science. The data extraction is developed by the reviewers based on JBI guidelines and relevant literature, the form will detail the characteristics of included publications, personal factors influencing cardiac rehabilitation engagement, and intervention characteristics. The data analysis will summarise descriptively the key features of the included studies and interventions, the Patient Health Engagement Model will guide the categorisation of personal factors into cognitive, emotional, and behavioural aspects, with other personal factors organised as emerging other relevant factors themes. The findings of this review will provide important evidence support for researchers, clinicians and policy makers to promote participation in cardiac rehabilitation. Within the constraints of medical and human resources, attention to personal factors can maximise the individual's role in cardiac rehabilitation and self-management, contributing to the efficient allocation and use of resources.

**Data availability statement:** No datasets were generated or analysed during the current study. All relevant data from this study will be made available upon study completion.

**Funding:** The author HZ has received the grant which Grant number is No. 202306370010 from China Scholarship Council. The funder's website is: https://www.csc.edu.cn/chuguo. The funders had no role in study design, data collection and analysis, decision to publish, or preparation of the manuscript.

**Competing interests:** The authors have declared that no competing interests exist.

# Introduction

Cardiovascular disease (CVD) is the leading cause of death and disability globally [1,2], accounting for 17.9 million deaths annually [3]. The overall global prevalence of CVD and the number of years lived with disability have nearly doubled in the last three decades [2]. Most cardiovascular diseases are chronic or recurrent, and their treatment and rehabilitation often follow individuals throughout their lives [4]. Therefore, individuals with cardiovascular disease have responsibility for the self-management of their condition and compliance and adherence to treatment and rehabilitation are critical to maintaining long-term health [5,6].

Cardiac rehabilitation (CR) is a comprehensive, evidence-based model of CVD care that includes exercise, education on modifiable risk factors and lifestyle changes, and psychological and social support [7]. Evidence suggests that cardiac rehabilitation can manage an individual's cardiac symptoms and improve their psychosocial status, effectively reduce the risk of readmission, and reduce mortality by 25% in a cost-effective manner [4,8]. The American Heart Association (AHA), the American College of Cardiology (ACC), and the European Society of Cardiology (ESC) have recommended cardiac rehabilitation in their clinical guidelines for cardiovascular disease [9–12].

Despite evidence supporting the effectiveness of cardiac rehabilitation, its utilisation remains suboptimal [6,13]. Available data suggest that in most countries, less than 50% of eligible individuals with cardiovascular disease enrol in cardiac rehabilitation [14]. In fact, Turk-Adawi et al. report that even when individuals with cardiovascular disease are enrolled in a cardiac rehabilitation programme, 56% to 82% of them do not adhere to or complete the programme across countries with different income levels [15]. The factors that influence individuals' engagement in cardiac rehabilitation in various aspects such as enrolment, adherence, and completion are different [16]. The main reason reported for individuals' non-enrolment in cardiac rehabilitation is a lack of referral to cardiac rehabilitation services [17]. Other reasons for non-enrolment may include insufficient awareness of the benefits of rehabilitation among individuals or healthcare providers, availability of cardiac rehabilitation services, lack of financial support, and missed optimal referral or enrolment time [18,19]. The factors affecting individuals adherence and completion of cardiac rehabilitation have been typically classified into three categories in previous studies, including personal factors (gender, age, health status, motivation, perceptions and beliefs about rehabilitation), healthcare provider factors (lack of knowledge about the benefits of cardiac rehabilitation), and healthcare system factors (timing and location of cardiac rehabilitation programmes, referrals) [20–25]. However, it has been found that promoting individuals' participation in cardiac rehabilitation by modifying factors that influence healthcare providers and the healthcare system is difficult, as these factors are heavily influenced by resource constraints such as funding and staffing [26–30]. There is a growing recognition that individuals' engagement in care is essential for enhancing health behaviours and clinical outcomes, and understanding and encouraging patients to actively participate in the healthcare process can improve efficiency and conserve healthcare resources [27,28,31]. Therefore, interventions to promote engagement in existing cardiac rehabilitation could be more feasible when designed to address modifiable personal factors [30,32,33]. Specifically, interventions could target the subjective experiential dimensions of modifiable personal factors, such as individuals' perceptions of their health status and treatment plans, as well as the emotional responses and psychological states experienced during the process of cardiovascular disease management [30,32,33].

Currently, interventions to promote engagement in cardiac rehabilitation are often designed to address multiple factors and some of these target subjective experiential

dimensions of personal factors. Examples include providing educational materials to increase individuals' knowledge of cardiac rehabilitation, using behaviour change techniques (e.g., goal setting, self-monitoring, and problem-solving strategies), and offering brief psychological interventions [6,30,32,34–36]. However, there is a lack of syntheses of the existing literature evidence on addressing personal factors that influence engagement in cardiac rehabilitation. Furthermore, existing reviews of factors and interventions influencing engagement in cardiac rehabilitation do not adequately concern the influence of individuals' subjective experiences and emotional changes on their engagement, which are critical for developing and maintaining intrinsic motivation for long-term recovery [23,37]. Therefore, a scoping review specifically focusing on the subjective experiential dimensions of personal factors influencing engagement in cardiac rehabilitation and mapping the body of evidence for the interventions targeting these dimensions is needed.

Person-centred healthcare has emerged as a significant trend in healthcare development over the past three decades, emphasising the treatment of individuals following a holistic care approach and encouraging their active participation in healthcare decisions [38,39]. Individuals' roles have become more diverse in the context of person-centred healthcare, where the healthcare system views patients as consumers of healthcare services with the "privilege" of choice and voice [40–42]. Based on this background, Paige et al. constructed a Patient Health Engagement (PHE) Model that conceptualizes the consumer psychology of patient engagement in healthcare [43,44]. The model defines patient health engagement as a multidimensional and psychosocial process resulting from the fully engaged individual's cognitive, emotional, and behavioural enactment of their health status and health management [44]. The model has been previously used to guide the exploration of factors influencing self-health management engagement in individuals with diabetes and heart failure [45,46]. Therefore, the Patient Health Engagement Model will provide a systematic perspective for this scoping review to explore personal factors that influence engagement of cardiac rehabilitation in terms of cognitive, emotional, and behavioural aspects [47].

The aim of this scoping review is to conduct a comprehensive review and mapping of the subjective experiential dimensions of personal factors and interventions targeting personal factors that facilitate or impede engagement in cardiac rehabilitation. The review will address the following review questions:

1. Which subjective experiential dimensions of personal factors influence cardiac rehabilitation engagement (enrolment, adherence, and completion) in individuals with cardiovascular diseases?

2. What are the interventions reported in the literature that aim at increasing cardiac rehabilitation engagement by addressing subjective experiential dimensions of personal factors?

## Materials and methods

The proposed scoping review will be conducted by following the Joanna Briggs Institute methodology for scoping reviews [48]. This comprises of the following six stages: (1) identifying the research question; (2) identifying relevant studies; (3) study selection; (4) charting the data; (5) collating, summarizing and reporting the results and (6) stakeholder consultation. This methodology was selected due to the broad scope of engagement in cardiac rehabilitation and the diverse, heterogeneous literature on personal factors, making a scoping review ideal for comprehensive mapping [48]. This scoping review will adhere to the PRISMA-P (Preferred Reporting Items for Systematic review and Meta-Analysis Protocols) checklist [49], which is provided as an online supplemental in S1 File. The scoping review is planned to begin in March 2025 and end by June 2025.

## Protocol and registration

This protocol has been registered at the Open Science Framework (https://osf.io/p8aes/).

## Eligibility criteria

**Publication type.** All types of peer-reviewed journal articles (e.g. quantitative, qualitative, and mixed-methods research designs, as well as reviews) will be included. Documents such as national recommendations, as well as grey literature including conference abstracts, dissertations, and blogs, will be excluded. Articles published in English will be included. Only literature from 2000 to the present will be included, due to substantial changes in the understanding of cardiac disease and cardiac rehabilitation over the course of the 21st century, leading to advances in cardiac rehabilitation intervention strategies [4,6].

**Population.** The population of the eligible articles will be adult population aged 18 or over with cardiovascular disease such as coronary heart disease, myocardial infarction, heart failure, valvular heart disease, or congenital heart disease. Publications reporting information about individuals from any gender and background will be included.

**Concept.** The publications must address subjective experiential dimensions of personal factors that influence engagement (enrolment, adherence, and completion) in cardiac rehabilitation, or report an intervention that is designed to increase engagement (enrolment, adherence, and completion) in cardiac rehabilitation by addressing these dimensions. Guided by the PHE model, this scoping review defines the subjective experiential dimensions of personal factors as comprising cognitive, emotional, and behavioral aspects of personal factors [43]. Cognition refers to an individual's ability to understand and reflect on their disease and its management, including knowledge of their health status, understanding of treatment plans, and ability to access and process health information [43]. Emotion refers to the emotional experiences of individuals when facing their disease and treatment, such as anxiety, fear, loss of control, or hope [43]. Behaviour refers to actions individuals take in managing their health, such as medication adherence, lifestyle changes, and actively seeking information and support [43]. The publications could be related to any form (e.g. tele-rehabilitation or home-based rehabilitation) or stages (e.g. acute, early convalescent and long-term maintenance) of cardiac rehabilitation.

**Context.** The definition, type and stages of cardiac rehabilitation in this review are based on the position paper of the Secondary Prevention and Rehabilitation Section of the European Association for Preventive Cardiology [12], which describes cardiac rehabilitation as a comprehensive approach that includes a variety of interventions such as medical assessment, exercise training, nutritional guidance, and psychological support [12]. There will be no restrictions on the setting of the study, we will include acute, hospital and community settings, as well as rehabilitation and transitional care units in rural and urban areas in different regions and countries.

## Information sources and search

A literature search will be conducted based on the three-step search strategy recommended by the JBI guidelines. First, an initial search will be conducted in the PubMed and CINAHL databases. This step has been partially completed by using the initial keywords including but not limited to: "Cardiovascular Disease", "Cardiac rehabilitation", "Personal factors", "Enrolment", "Adherence", "Completion", "Engagement", "Barriers", "Facilitators", "Psychological factors", "Motivational factors", "intervention". Following this, the text words contained in the titles and abstracts of the retrieved papers and the index terms used to describe the articles will be analysed. Afterwards, a second search using all identified keywords and index terms

across all selected databases will be conducted. Reference tracking of included articles will be conducted if any potentially relevant publications are identified. Where needed, the reviewers will proactively contact the authors for further information.

In collaboration with a librarian, tailored search strategies for different databases to retrieve literature will be developed, using PubMed, Embase, Cochrane Library, CINAHL, PsycINFO, Scopus, and Web of Science. Google Scholar will be also searched for additional articles. An example of the pilot search strategy for PubMed is provided in online supplemental S2 File.

## Selection of sources of evidence

An online platform, Rayyan, will be used for study selection. Prior to this, the records will be de-duplicated using Endnote and Rayyan. The study selection will be conducted through a two-step screening process, starting with title and abstract screening, followed by a full-text review [50]. Initially, two independent reviewers (HZ and ZW) will screen the titles and abstracts based on the pre-defined eligibility criteria. Any abstract or title deemed relevant by both reviewers will advance to the full-text review stage. A PRISMA flow diagram (S3 File) will be included in the final publication to illustrate the results at each review stage [49]. Discrepancies between the reviewers will be resolved through discussion, and if consensus cannot be reached, a third reviewer (DS) will be consulted. After the title and abstract screening and resolution of conflicts, the first reviewer (HZ) will screen the full texts of the selected articles to determine their eligibility for inclusion. Before this, 10% of the articles will be screened by two reviewers (HZ and DS) to ensure that inclusion and exclusion criteria are applied consistently. Reasons for the exclusion of ineligible articles will be thoroughly documented.

## Data charting process and data items

The data extraction will be conducted by the first reviewer (HZ). The data extraction form will be piloted and discussed by two reviewers (HZ and DS) before the data extraction commences. The data extraction form was developed by the reviewers following the JBI guidance and the relevant literature [50]. Data extracted from each article will include study characteristics (authors, year, country, design, setting, purpose, theoretical framework, stage of cardiac rehabilitation, type of cardiac rehabilitation, and participants), personal factors influencing engagement in cardiac rehabilitation, and specific details of the intervention (purpose, theoretical basis, components, mode of delivery, implementing professionals' background, duration, location, and targeted individual factors). The data extraction for subjective experiential dimensions of personal factors will be based on the definitions of the three aspects outlined in the PHE model (cognitive, emotional, and behavioural). However, any additional personal factors reported in the included articles will also be considered as other relevant factors [46]. The data extraction form draft can be found in S4 File. The draft data extraction form will be modified and revised as needed during the pilot extraction process and all modifications will be documented in the scoping review.

## Critical appraisal of individual sources of evidence

A quality assessment of the included studies will not be conducted, as this is not mandatory for scoping reviews.

## Synthesis of results

The data synthesis for this review will include a mapping of the data and a narrative synthesis. The main characteristics of the included studies will be summarised descriptively,

highlighting the publication type, participant characteristics, locations, and other descriptors of the data items. Given the various forms of cardiac rehabilitation—such as centre-based, home-based, and remote—and the potential variation in personal factors across these forms, we will first categorise the different types of cardiac rehabilitation. Subsequently, personal factors for cardiac rehabilitation engagement will be mapped to the various forms of cardiac rehabilitation and narratively summarised with the guidance of the Patient Health Engagement Model, which provides a framework for mapping individual factors and intervention components across the three dimensions of cognition, emotion, and behaviour [43]. The intervention types and components will also be categorised and presented as narrative summaries.

## Discussion

This scoping review will be conducted to identify the available evidence on the subjective experiential dimensions of personal factors that influence engagement in cardiac rehabilitation and explore interventions targeting personal factors to improve engagement in cardiac rehabilitation. It will be guided by the Patient Engagement Model when identifying the personal factors with a specific focus on the influence of individuals' emotions, cognitive and behavioural processes. This scoping review will examine the variations in personal factors associated with different forms of cardiac rehabilitation, placing particular emphasis on analysing the distinctions between these factors across various rehabilitation modalities. To ensure the reproducibility of the study, a detailed plan for conducting the review was outlined in this protocol, and any changes to the review protocol will be reported in the scoping review. The potential limitations of this scoping review include language restrictions and the exclusion of grey literature due to the constrains with time and resources. Despite these limitations, to our knowledge, this is the first scoping review that comprehensively explores personal factors associated with cardiac rehabilitation engagement. The findings of this review will provide significant evidence to support researchers, clinicians, and policymakers in promoting cardiac rehabilitation engagement. Given the constraints of medical and human resources, focusing on personal factors can maximise individual agency, ensure efficient use of limited resources, and leverage individuals' potential in long-term rehabilitation and self-management.

## Supporting information

**S1 File. Preferred Reporting Items for Systematic reviews and Meta-Analyses extension for Scoping Reviews (PRISMA-ScR) Checklist.**
(DOCX)

**S2 File. Sample search strategy for PubMed.**
(DOCX)

**S3 File. PRISMA flow diagram.**
(DOCX)

**S4 File. Data extraction form.**
(DOCX)

## Acknowledgments

We thank the University of Galway Library staff members for their contributions to the development of database search strategies. Moreover, we thank Ziyue Wang (ZW) for agreeing to support the work of title and abstract screening process for this scoping review.

## Author contributions

**Conceptualization:** Hongyu Zhang, Duygu Sezgin.

**Writing – original draft:** Hongyu Zhang.

**Writing – review & editing:** Duygu Sezgin.

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
