## [Decision Letter · Decision Letter 0]

25 Nov 2024

PONE-D-24-30794Identification of personal factors that influence engagement in cardiac rehabilitation and interventions targeting personal factors: a scoping review protocolPLOS ONE

Dear Dr. Zhang,

Thank you for submitting your manuscript to PLOS ONE. After careful consideration, we feel that it has merit but does not fully meet PLOS ONE’s publication criteria as it currently stands. Therefore, we invite you to submit a revised version of the manuscript that addresses the points raised during the review process.

We look forward to receiving your revised manuscript.

Kind regards,

Eisuke Amiya

Academic Editor

PLOS ONE

Journal requirements:    When submitting your revision, we need you to address these additional requirements. 1. Please ensure that your manuscript meets PLOS ONE's style requirements, including those for file naming. The PLOS ONE style templates can be found at https://journals.plos.org/plosone/s/file?id=wjVg/PLOSOne_formatting_sample_main_body.pdf and https://journals.plos.org/plosone/s/file?id=ba62/PLOSOne_formatting_sample_title_authors_affiliations.pdf

Additional Editor Comments:

This is an important issue to enhance the impact of cardiac rehabilition for patients with cardiovascular disease.

However, there were some points to be addressed.

#This review focuses on the association between the participation in cardiac rehabilitation and the patient's personal factors. However, cardiac rehabilitation has many different forms such as center-based, home-based and remote, and the barriers to participation vary depending on the form. It would be desirable to consider this point as well, and analysis is also required regarding the differences in association depending on the form.

# "Personal factor" is an extremely vague term. Does it only include psychological tendencies?

If multiple factors such as economic situation, education, living environment, and work were included, it would be expected that a more multifaceted examination would be possible.

It is also necessary to consider whether screening using the word "personal factors" can sufficiently gather relevant papers.

Reviewers' comments:

Reviewer's Responses to Questions

**Comments to the Author**

1. Does the manuscript provide a valid rationale for the proposed study, with clearly identified and justified research questions?

Reviewer #1: Yes

2. Is the protocol technically sound and planned in a manner that will lead to a meaningful outcome and allow testing the stated hypotheses?

Reviewer #1: Yes

3. Is the methodology feasible and described in sufficient detail to allow the work to be replicable?

Reviewer #1: Yes

4. Have the authors described where all data underlying the findings will be made available when the study is complete?

Reviewer #1: Yes

5. Is the manuscript presented in an intelligible fashion and written in standard English?

Reviewer #1: Yes

6. Review Comments to the Author

You may also provide optional suggestions and comments to authors that they might find helpful in planning their study.

Reviewer #1: this scoping review protocol is clearly and fully described. Background and rationale, aims, methods, expected results and conclusions that can be drawn are depicted and appropriate.

no further questions or comments. looking forward for the results.

7. PLOS authors have the option to publish the peer review history of their article (what does this mean? ). If published, this will include your full peer review and any attached files.

**Do you want your identity to be public for this peer review?** For information about this choice, including consent withdrawal, please see our Privacy Policy .

Reviewer #1: **Yes: ** Laura Adelaide Dalla Vecchia

---

## [Author Response · Author response to Decision Letter 0]

20 Dec 2024

Editor comments

This is an important issue to enhance the impact of cardiac rehabilition for patients with cardiovascular disease. However, there were some points to be addressed.

Response: We greatly appreciate the efforts that you paid to review this manuscript. The editor and reviewer’s comments are extremely helpful.The manuscript has been revised accordingly. We have marked the revisions in red in the revised manuscript.

Comment 1: This review focuses on the association between the participation in cardiac rehabilitation and the patient's personal factors. However, cardiac rehabilitation has many different forms such as center-based, home-based and remote, and the barriers to participation vary depending on the form. It would be desirable to consider this point as well, and analysis is also required regarding the differences in association depending on the form.

Response: Thank you for your advice and guidance. We had previously considered that different forms of cardiac rehabilitation may be influenced by varying personal factors, and therefore, we mentioned the inclusion of different forms of cardiac rehabilitation in the inclusion and exclusion criteria, search strategy, and data extraction sections of the paper:

1.Considering that different forms of cardiac rehabilitation may be influenced by various personal factors, we have specified the following in the inclusion and exclusion criteria: “The publications could be related to any form (e.g. tele-rehabilitation or home-based rehabilitation)” (Please see lines 176-177 on page 9).

2.In the search strategy, we did not restrict the types of cardiac rehabilitation to ensure that all relevant literature on different types of cardiac rehabilitation could be identified (Please see Appendix S2).

3.In the Data Extraction Table, we planned to extract the type of cardiac rehabilitation to assist with subsequent analysis (Please see Appendix S4).

We have revised the text as follows to address your concerns and hope that it is now clearer:

“Synthesis of Results:

‘Given the various forms of cardiac rehabilitation—such as centre-based, home-based, and remote—and the potential variation in personal factors across these forms, we will first categorise the different types of cardiac rehabilitation. Subsequently, personal factors for cardiac rehabilitation engagement will be mapped to the various forms of cardiac rehabilitation and narratively summarised with the guidance of the Patient Health Engagement Model.’

(Please see lines 249–255 on page 12-13 )

Discussion:

‘This scoping review will examine the variations in personal factors associated with different forms of cardiac rehabilitation, placing particular emphasis on analysing the distinctions between these factors across various rehabilitation modalities.’

(Please see lines 265–268 on page 13 )”

Comment 2: "Personal factor" is an extremely vague term. Does it only include psychological tendencies? If multiple factors such as economic situation, education, living environment, and work were included, it would be expected that a more multifaceted examination would be possible.

Response: Thank you for your advice. This scoping review focuses on the subjective experiential dimensions of personal factors, guided by the Patient Health Engagement (PHE) model, to explore cognitive, emotional, and behavioral aspects of personal factors. Furthermore, if the included literature identifies other personal factors beyond these three aspects, they will be considered as other relevant factors. Guided by the PHE model, this scoping review defines the subjective experiential dimensions of personal factors as comprising cognitive, emotional, and behavioral aspects of personal factors [43]. Cognition refers to an individual’s ability to understand and reflect on their disease and its management, including knowledge of their health status, understanding of treatment plans, and ability to access and process health information [43]. Emotion refers to the emotional experiences of individuals when facing their disease and treatment, such as anxiety, fear, loss of control, or hope [43]. Behaviour refers to actions individuals take in managing their health, such as medication adherence, lifestyle changes, and actively seeking information and support [43]. (Please see lines 168-177 on page 9)

This is designed based on the following considerations:

1.As discussed in the background section of the manuscript, despite the growing interest in cardiac rehabilitation participation and the increasing discussions around strategies to enhance engagement, the psychological and emotional experiences of individuals actively participating in healthcare processes have been largely overlooked. Our research aims to address this gap and contribute to a deeper understanding of cardiac rehabilitation participation.

2.We agree that the term “personal factors” is extremely vague. Looking at all possible personal factors would result in an overly broad review scope, making it difficult to derive focused and specific conclusions. The PHE model provides a framework that specifically and accurately focuses on the subjective experiential dimensions of individuals' participation in care processes. Therefore, we have narrowed down the scope of “personal factors” to “subjective experiential dimensions”, ensuring the review remains focused and aligned with its core objectives, thereby enhancing its depth and relevance. To avoid missing important information, we will also document and analyse any other personal factors reported in the included literature.

To better clarify the concept of “personal factors”, we made the following revisions in the manuscript:

“Introduction:

‘Specifically, interventions could target the subjective experiential dimensions of modifiable personal factors, such as individuals' perceptions of their health status and treatment plans, as well as the emotional responses and psychological states experienced during the process of cardiovascular disease management [30, 32, 33].’

(Please see lines 86-90 on page 3)

‘Currently, interventions to promote engagement in cardiac rehabilitation are often designed to address multiple factors and some of these target subjective experiential dimensions of personal factors. Examples include providing educational materials to increase individuals' knowledge of cardiac rehabilitation, using behaviour change techniques (e.g., goal setting, self-monitoring, and problem-solving strategies), and offering brief psychological interventions [6, 30, 32, 34-36].’

(Please see lines 91-96 on page 5)

‘Therefore, a scoping review specifically focusing on the subjective experiential dimensions of personal factors influencing engagement in cardiac rehabilitation and mapping the body of evidence for the interventions targeting these dimensions is needed.’

(Please see lines 102-105 on page 6)

‘The aim of this scoping review is to conduct a comprehensive review and mapping of the subjective experiential dimensions of personal factors and interventions targeting personal factors that facilitate or impede engagement in cardiac rehabilitation. The review will address the following review questions:

1.Which subjective experiential dimensions of personal factors influence cardiac rehabilitation engagement (enrolment, adherence, and completion) in individuals with cardiovascular diseases?’

2.What are the interventions reported in the literature that aim at increasing cardiac rehabilitation engagement by addressing subjective experiential dimensions of personal factors?’

(Please see lines 123-132 on page 7)

Eligibility criteria:

‘The publications must address subjective experiential dimensions of personal factors that influence engagement (enrolment, adherence, and completion) in cardiac rehabilitation, or report an intervention that is designed to increase engagement (enrolment, adherence, and completion) in cardiac rehabilitation by addressing these dimensions. Guided by the PHE model, this scoping review defines the subjective experiential dimensions of personal factors as comprising cognitive, emotional, and behavioral aspects of personal factors [43]. Cognition refers to an individual’s ability to understand and reflect on their disease and its management, including knowledge of their health status, understanding of treatment plans, and ability to access and process health information [43]. Emotion refers to the emotional experiences of individuals when facing their disease and treatment, such as anxiety, fear, loss of control, or hope [43]. Behaviour refers to actions individuals take in managing their health, such as medication adherence, lifestyle changes, and actively seeking information and support [43].’

(Please see lines 164-177 on page 8-9)

Data charting process and data items:

‘The data extraction for subjective experiential dimensions of personal factors will be based on the definitions of the three aspects outlined in the PHE model (cognitive, emotional, and behavioural). However, any additional personal factors reported in the included articles will also be considered as other relevant factors [46].’

(Please see lines 234-238 on page 12)”

Comment 3: It is also necessary to consider whether screening using the word "personal factors" can sufficiently gather relevant papers.

Response: Thank you for your advice and guidance.

Based on the discussions with the librarian and results from the pilot search, we did not use the word "personal factors" directly as a search term, as it might limit the scope and number of search results. Instead, we followed a broad approach and used the search terms related to patient participation and engagement in cardiac rehabilitation as outlined in Appendix S2 for our initial search strategy. In subsequent work, we will continue to refine the search strategy as needed to ensure more relevant and comprehensive search results.

Reviewer #1 comments

This scoping review protocol is clearly and fully described. Background and rationale, aims, methods, expected results and conclusions that can be drawn are depicted and appropriate.

No further questions or comments. looking forward for the results.

Response: Thank you very much for your feedback.

---

## [Decision Letter · Decision Letter 1]

14 Jan 2025

Identification of personal factors that influence engagement in cardiac rehabilitation and interventions targeting personal factors: a scoping review protocol

PONE-D-24-30794R1

Dear Dr. Zhang,

We’re pleased to inform you that your manuscript has been judged scientifically suitable for publication and will be formally accepted for publication once it meets all outstanding technical requirements.

Kind regards,

Eisuke Amiya

Academic Editor

PLOS ONE

Additional Editor Comments (optional):

Reviewers' comments:

Reviewer's Responses to Questions

**Comments to the Author**

1. Does the manuscript provide a valid rationale for the proposed study, with clearly identified and justified research questions?

Reviewer #1: Yes

Reviewer #2: Yes

Reviewer #3: Yes

2. Is the protocol technically sound and planned in a manner that will lead to a meaningful outcome and allow testing the stated hypotheses?

Reviewer #1: Yes

Reviewer #2: Yes

Reviewer #3: Yes

3. Is the methodology feasible and described in sufficient detail to allow the work to be replicable?

Reviewer #1: Yes

Reviewer #2: Yes

Reviewer #3: Yes

4. Have the authors described where all data underlying the findings will be made available when the study is complete?

Reviewer #1: Yes

Reviewer #2: Yes

Reviewer #3: Yes

5. Is the manuscript presented in an intelligible fashion and written in standard English?

Reviewer #1: Yes

Reviewer #2: Yes

Reviewer #3: Yes

6. Review Comments to the Author

You may also provide optional suggestions and comments to authors that they might find helpful in planning their study.

Reviewer #1: I have no further questions. The authors' replies are accetable and the manuscript has been improved

Reviewer #2: Identification of personal factors that influence engagement in cardiac rehabilitation and interventions targeting personal factors: a scoping review protocol

The authors addressed the given comments.

Reviewer #3: This protocol manuscript is well-written and described. No additional comments or questions are needed

7. PLOS authors have the option to publish the peer review history of their article (what does this mean? ). If published, this will include your full peer review and any attached files.

**Do you want your identity to be public for this peer review?** For information about this choice, including consent withdrawal, please see our Privacy Policy .

Reviewer #1: **Yes: ** Laura Adelaide Dalla Vecchia

Reviewer #2: No

Reviewer #3: No

---

## [Editor Report · Acceptance letter]

PONE-D-24-30794R1

PLOS ONE

Dear Dr. Zhang,

I'm pleased to inform you that your manuscript has been deemed suitable for publication in PLOS ONE. Congratulations! Your manuscript is now being handed over to our production team.

Kind regards,

on behalf of

Dr. Eisuke Amiya

Academic Editor

PLOS ONE